# LOCAL SUPERIOR SOUPS: A CATALYST FOR REDUCING COMMUNICATION ROUNDS IN FEDERATED LEARNING WITH PRE-TRAINED MODEL

## ABSTRACT

Federated learning (FL) is a learning paradigm that enables collaborative training of models using decentralized data. Recently, the utilization of pre-trained weight initialization in FL has been demonstrated to effectively improve model performance. However, the current pre-trained models have become increasingly parameter-rich. The sheer scale of model parameters introduces substantial communication rounds challenges during their adaptation to FL. To address these communication cost issues and elevate the performance of pre-trained model adaptation in FL, we propose an innovative model interpolation-based local training technique called "Local Superior Soups." Our method promotes local training across different clients, encouraging the exploration of a connected low-loss basin within a few communication rounds through regularized model interpolation. This approach serves as a facilitator for pre-trained model adaptation in FL. We demonstrated its effectiveness and efficiency across diverse widely-used FL datasets.

## 1 INTRODUCTION

Federated Learning (FL) (McMahan et al., 2017) has emerged as a promising methodology for leveraging the power of private data without the need for centralized data governance. However, data heterogeneity in FL poses significant challenges to the design of efficient training for global convergence. With the emergence of the pre-training and fine-tuning paradigm in various applications (He et al., 2019; Hu et al., 2022), recent studies (Nguyen et al., 2022; Chen et al., 2022) have attempted to address the problem of FL under data heterogeneity with pre-trained initialization. Although pre-trained federated learning can speed up convergence compared to random initialization, it still requires a significant number of communication rounds between the server and clients, often amounting to hundreds of rounds (Nguyen et al., 2022). Existing pre-trained models Radford et al. (2021); Touvron et al. (2023) often have an enormous parameter scale, and following the neural scaling law (Kaplan et al., 2020), there is a continuous trend toward increasing model parameters. Deploying models with such a large parameter size in FL introduces significant communication overhead. This greatly hampers the flexibility and scalability of model updates. Reducing FL communication overhead can be approached by reducing the scale of model parameters involved in distributed training (Zhang et al., 2023) or reducing communication rounds (McMahan et al., 2017). Comparing with reducing model parameters, reducing communication rounds typically leads to a more efficient reduction of network congestion (Hegde et al., 2023), decreased energy consumption on client devices (Luo et al., 2021), and a lower risk of privacy breaches (Zhu et al., 2019). *In this paper, we focus on reducing communication rounds in FL with pre-trained model as initialization.*

Typically, increasing the number of local training steps can effectively reduce communication rounds. However, there is an upper limit to the extent of local training step increments. This limitation arises due to the presence of data heterogeneity, where the optimization consistency among different clients deteriorates with the increasing number of local steps. This optimization inconsistency leads to discrepancy between local and global models and decelerates the convergence rate of FL. The discrepancy is often referred to as client drift (Karimireddy et al., 2020). Previously, some FL methods (Karimireddy et al., 2020; Sun et al., 2023) attempted to introduce proximal terms to regularize local training, aiming to reduce local overfitting and mitigate the issue of client drift. While these methods can

accelerate convergence, they restrict the progress of each local training steps towards the optimal solution, impeding the attainment of FL with more aggressive communication round reductions.

Furthermore, these client drift mitigation methods can alleviate local overfitting to some extent but still cannot guarantee that the globally aggregated models perform well. This situation arises when individual local clients become trapped in isolated low-loss valleys. More specifically, as illustrated in Figure 1, two models from clients 'A' and 'B', even if their optimal model distance is small, still result in a poorly performing aggre-

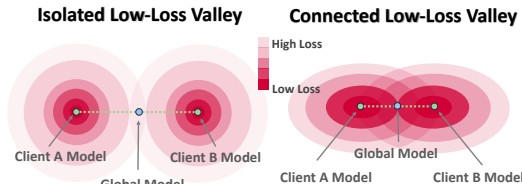

Figure 1: Illustration on isolated (left) and connected low-loss valley with larger regions in dark red (right).

gated global model. Moreover, the preceding FL methods aimed at minimizing communication rounds exclusively address scenarios involving random initialization, lacking a customized approach tailored to pre-trained models. Recent proposed centralized fine-tuning methods (*e.g.*, model soups (Wortsman et al., 2022) and DiWA (Ramé et al., 2022b) – a greedy model selection version of model soups) based on model interpolation (averages of a large number of model weights) are effective approaches to seek large connected low-loss region, which are promising for applying in FL to reduce communication rounds. These methods can prevent individual clients from being trapped in isolated low-loss valleys by positioning global model centrally within a larger low-loss region by overlapping the low-loss regions among clients, as shown in Fig. 1 (right). However, their training efficiency is exceedingly low, requiring the complete retraining of numerous models, leading to *significant computational overhead* on clients and intolerable communication costs when applied in FL, due to two aspects: Firstly, they involve a time-consuming model selection phase within the model pool, which consists of all candidate models available for weight interpolation. Secondly, model soups entail an extensive number of model training iterations, lacking prior guidance and relying on brute-force, random, and often redundant training. Many of the trained models end up unused.

To enjoy the connected low-loss valley benefits of model soups-based methods (Wortsman et al., 2022; Ramé et al., 2022b) without burdening local training, we propose an efficient and local model interpolation-based method, called **Local Superior Soups** (**LSS**). To address the first issue, we propose a **sequential random model interpolation** method during training. This eliminates the need for subsequent model selection steps and ensures that the models slated for interpolation reside within the same low-loss valley during training (Sec. 3.3.1). For the second issue, we introduce two quantifiable indicators of candidate model quality: **diversity** (Sec. 3.3.2) and **affinity** (Sec. 3.3.3). Specifically, the *diversity indicator* quantifies the diversity among models in the model pool with their pairwise model distance, where larger distances denote higher diversity, signifying better model quality. As illustrated in Figure 2 (left), a low-loss region can be effectively covered with only a few candidate models positioned near its periphery. Thus, we propose incorporating diversity metric as a regularization term during training to maximize the expansion of low-loss regions, thereby increasing the utilization of trained models.

The *affinity indicator* measures the affinity of each candidate model in the model pool to the initial model. Smaller distances indicate greater affinity, indicating better model quality. This affinity is also incorporated as a regularization term during training to prevent the expansion of low-loss regions from deviating too far from the shared initialization point, thus reducing the likelihood of overlapping connected regions (as depicted in right side of Fig. 2). These two indicators facilitate the efficient inclusion of models into the model pool, preventing the wasteful

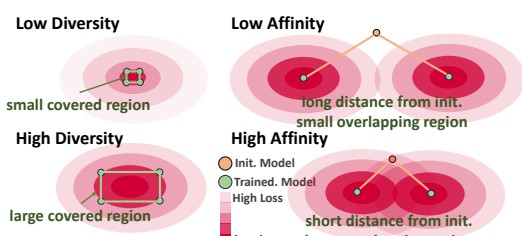

Figure 2: Illustration on diversity (left) and affinity (right) regularization.

training of models that may ultimately go unused. In experiments, we found that our proposed method greatly reduces communication rounds, and we achieved the performance of models fused after multiple rounds of communication in other FL methods with only a few rounds of communication.

In summary, our contributions are as follows:

(1) We reveal the importance of regularizing local client models in the connected low-loss valleys for reducing communication rounds when initializing FL with pre-trained models. (2) We introduce an innovative and efficient model soups-based method for FL, called Local Superior Soups (*LSS*) that eliminates the need for time-consuming model selection and redundant model training in the existing soups-based approaches, while expanding connected low-loss valleys of client models for faster convergence. (3) In experimental evaluations, *LSS* demonstrates a significant reduction in communication rounds, achieving superior performance with only a few rounds of communication, exceeding baseline FL methods significantly in four datasets and two types of distribution shifts.

## 2 RELATED WORK

### 2.1 HETEROGENEOUS FEDERATED LEARNING

FL struggles with Non-IID data, leading to various proposed algorithms. FedProx (Li et al., 2020) uses proximal term to regularize local training, preventing client divergence. Scaffold (Karimireddy et al., 2020) adds variance reduction to combat "clients-drift." MOON (Li et al., 2021a) employs mode-level contrastive learning to stabilize local training. Personalized FL (Tan et al., 2021) targets high local performance on Non-IID data. FedBN (Li et al., 2021b) applies local batch normalization to mitigate feature shift before model averaging. Recent one-shot communication round FL methods utilize server-side techniques like prediction ensembles (Guha et al., 2019) or generating data (Zhang et al., 2022a; Heinbaugh et al., 2023) for centralized training, improving aggregated model performance. Few-round communication round FL, based on meta-learning (Park et al., 2021), may not align with practical FL scenarios due to data partition concerns.

### 2.2 FINE-TUNING AND MODEL INTERPOLATION

Fine-tuning leverages pre-trained models to enhance task performance (Choshen et al., 2022). FedFTG (Zhang et al., 2022b) proposes knowledge distillation for global model fine-tuning in FL. Personalized FL employs fine-tuning to adapt global models to local ones, e.g., FedBABU (Oh et al., 2022), FTFA, and RTFA (Cheng et al., 2021). However, this focus on local performance neglects global generalization. Inspired by linear mode connectivity (Nagarajan & Kolter, 2019; Frankle et al., 2020), Model Soups (Wortsman et al., 2022) combines runs with varied hyper-parameters. DiWA (Ramé et al., 2022b) extends this concept, emphasizing diverse hyper-parameter training. Soups-based methods (Wortsman et al., 2022; Ramé et al., 2022b) aggregate diverse models for better generalizability. Some methods induce diversity through high learning rates (Maddox et al., 2019), cosine similarity minimization (Wortsman et al., 2021), tempered posteriors (Izmailov et al., 2019), or auxiliary dataset-trained model soups (Ramé et al., 2022a). We depict the difference of different model ensemble-based methods in our appendix 7.

## 3 METHOD

The structure of this Section is as follows: firstly, we provide the problem definition and corresponding notions to be used (Sec. 3.1); secondly, we reveal the dilemma for existing federated learning methods on reducing communication rounds (Sec. 3.2); finally, we propose a regularized model interpolation-based method as a solution, provide corresponding analysis (Sec. 3.3), and present the overall algorithm flow.

### 3.1 NOTIONS AND PROBLEM DEFINITION

**Notions.** Let $\mathcal{X}$ be the input space of data, $\mathcal{Y}$ be the label space. Consider a FL setting with $M$ clients, $\tau$ local steps and $R$ communication rounds. Let $\mathcal{D} := \{\mathcal{D}_i\}_{i=1}^M$ be a set of $M$ training domain, each of which is a distribution over the input space $\mathcal{X}$. For each client, we have access to $n$ training data points in the form of $\mathcal{D}_i = (x_j^i, y_j^i)_{j=1}^n \sim \mathcal{D}_i$, where $y_j^i$ denotes the target label for input $x_j^i$. Let $f \in \mathbb{R}^m$ represents the parameter for the global model, $\ell_i : \mathbb{R}^m \to \mathbb{R}$ denotes the local objective function at client $i$, and $\mathcal{P}$ denotes a distribution on the entire set of clients.

**Problem definition.** We aim to address the challenge of optimizing the global performance on $\mathcal{D}$ of aggregated models fine-tuned from different clients with data heterogeneity, while minimizing

communication rounds between the clients and the server in data Non-IID setting. In terms of global performance, we perform empirical risk minimization (ERM) on the sampled data $\mathcal{D}_i$ for $i \in [M]$,

$$\mathcal{L}(f) = \sum_{i=1}^{M} p_i \mathcal{L}_i(f), \quad \text{where } \mathcal{L}_i(f) = \frac{1}{|\mathcal{D}_i|} \sum_{\xi \in \mathcal{D}_i} \ell_i(f, \xi) \text{ and } \sum_{i=1}^{M} p_i = 1. \quad (1)$$

### 3.2 LOCAL STEPS EFFECT ON FL CONVERGENCE UNDER DATA HETEROGENEITY

In this section, we present the tool to understand how communication rounds and data heterogeneity affect the FL convergence bound. Formally, we present the error term and posit the following assumptions for the purpose of analysis.

Our analysis follows the assumptions and convergence bound in (Wang et al., 2021b), which is restated as follows. The formal statements are detailed in Appedinx B.1.

**Theorem 3.1** (Convergence Rate for Convex Local Functions, Theorem 1 in Wang et al. (2021b)). *Under Convexity and Smoothness Assumption on $\beta$-smooth loss function, Bounded Variance of Stochastic Gradient and Bounded Variance of Local and Global Gradient assumptions, when the client learning rate is chosen properly, we have*

$$\mathbb{E}\left[ \frac{1}{\tau R} \sum_{r=0}^{R-1} \sum_{k=1}^{\tau} \mathcal{L}(\overline{f}^{(r,k)}) - \mathcal{L}(f^\star) \right] \leq \frac{2\beta d^2}{\tau R} + \frac{2\sigma d}{\sqrt{M\tau R}} + \underbrace{\frac{5\beta^{\frac{1}{3}}\sigma^{\frac{2}{3}}d^{\frac{4}{3}}}{\tau^{\frac{1}{3}}R^{\frac{2}{3}}} + \frac{19\beta^{\frac{1}{3}}\zeta^{\frac{2}{3}}d^{\frac{4}{3}}}{R^{\frac{2}{3}}}}_{\text{errors from local updates \& Non-IID data}}.$$

$$(2)$$

Here, $d := \|f^{(0,0)} - f^\star\|$ refers to the distance between initialization $f^{(0,0)}$ and the global optimum $f^\star$, $\sigma$ bounds variance of stochastic gradient by $\mathbb{E}[\|g_i(f^{(r,k)}) - \nabla\mathcal{L}_i(f^{(r,k)})\|^2|f^{(r,k)}] \leq \sigma^2$, and $\zeta$ bounds variance of local and global gradient by $\max_i \sup_f \left\|\nabla\mathcal{L}_i(f^{(r,k)}) - \nabla\mathcal{L}(f^{(r,k)})\right\| \leq \zeta$.

**How to reduce communication rounds under data heterogeneity?** Increasing local fine-tuning steps seems to be a straightforward technique to reduce communication costs. Nevertheless, this approach cannot reduce the an error term in the convergence rate (see the 4th term of the RHS of Eq. 2), which remains unaltered by increasing local steps. Moreover, increasing local update steps in the presence of Non-IID client data exacerbates the inconsistency in local objectives, further magnifying this error term. Here, we provide a more detailed explanation, specifically identifying the conditions under which increasing iteration steps can effectively reduce communication rounds.

**Proposition 3.1.** *Under the data heterogeneity setting, when the total number of gradient computations across all clients ($K = M\tau R$) is fixed and the local steps $\tau$ satisfies*

$$\tau \leq \frac{\sigma}{\zeta} \sqrt{\frac{\sigma}{d\beta} \frac{K^{\frac{1}{2}}}{M^2}}, \quad (3)$$

*the error upper bound Eq.2 will be dominated by the second term $\mathcal{O}(1/\sqrt{K})$.*

We provide the proof for Proposition 3.1 in Appendix B.2. Accordingly, increasing the bound in Eq. 11 and meeting the aforementioned condition for local steps allows us to reduce communication rounds. From the above inequation, we can observe that although increasing the number of local training steps can reduce communication rounds, there is a limit to the number of steps that can be added. This limit is primarily determined by the error term introduced by local updates.

**Why connected low-loss valley + pre-trained initialization can achieve extreme communication rounds reduction?** Our analysis indicates that for substantial communication efficiency in federated learning, it is not enough to just increase local training steps. The focus should be on minimizing the error term from local updates, particularly the last term in Formula 2. This term, influenced by gradient dissimilarity ($\zeta$), distance to optimal weights ($d$), and the Lipschitz constant ($\beta$), remains significant even as training steps increase. Prior research suggests (Nguyen et al., 2022) that pre-training initialization reduces $\zeta$ by aligning client updates, and overparameterized models typically position the optimal solution close to the initialization point (Chizat et al., 2019; Li & Banerjee,

2021), decreasing $d$. And we argue that local training that connects low-loss regions can reduce $\beta$. Consequently, a combination of pre-training and our proposed connectivity preserving local training can effectively lower error terms from local updates, increasing the limit of local training steps and thus reducing communication rounds. More experimental support see our Appendix.

---

**Algorithm 1:** *LSS* (Local Training) Pseudo-code

---

**Require:** $f_p$ pre-trained model ($R = 1$) or global aggregated model ($R > 1$); $\mathcal{L}$ loss function; $\mathcal{D}$ dataset; $dist$ distance function; $\tau$ iteration steps; $\eta$ learning rate; $\lambda_a$ affinity coefficient; $\lambda_d$ diversity coefficient; $n$ number of averaged models.

1: ***LSS* Local Training :**
2: $\mathcal{M} \leftarrow \{f_p\}$
3: **for** $p_i = 1$ to $n$ **do**
4:     $f_{p_i} \leftarrow Averaging(\mathcal{M})$
5:     $\mathcal{M} \leftarrow \mathcal{M} \cup \{f_{p_i}\}$                         # sequential training with newly added model
6:     **for** $t = 1$ to $\tau$ **do**
7:         $f_s \leftarrow RandomInterpolation(\mathcal{M})$        # connectivity preserving
8:         $\mathcal{L}_{reg}(f_{p_i}) = \mathcal{L}(f_s, \mathcal{D}) + \lambda_a \cdot dist(f_{p_i}, f_p) - \lambda_d \cdot dist(f_{p_i}, \mathcal{M})$
9:         $f_{p_i} \leftarrow f_{p_i} - \eta \nabla_{f_{p_i}} \mathcal{L}_{reg}(f_{p_i})$
10: **Inference:**
11: $f \leftarrow Averaging(\mathcal{M})$

---

### 3.3 OUR SOLUTION: *LSS* ALGORITHM

In this part, we first present the shortcomings of the previous model soups method applied in FL. Secondly, we propose our three targeted improvements, *i.e.* **random model interpolation** (Sec. 3.3.1), **diversity term** (Sec. 3.3.2), and **affinity regularization term** (Sec. 3.3.3). Finally, we present the complete algorithm process and detailed implementation in local client training.

**Limitation of previous model soups methods.** Previous model soups methods (Wortsman et al., 2022) can induce a trained model located in a connected low-loss valley, but their training efficiency is exceedingly low, due to two aspects: *Time-Consuming model selection phase:* Firstly, these methods involve a time-consuming model selection phase, which consists of all candidate models available for weight interpolation. This phase aims to choose models that reside within the same low-loss valleys. During this selection process, significant computational resources are consumed to identify suitable models for interpolation, adding to the overall training time and complexity. *Extensive and redundant model training:* Secondly, model soups entail an extensive number of model training iterations, lacking prior guidance and relying on brute-force, random, and often redundant training. Many of the trained models end up unused, further exacerbating the computational inefficiency.

#### 3.3.1 RANDOM INTERPOLATION CONSERVING CONNECTED LOW-LOSS REGION.

To address the *time-consuming model selection* issue of the previous soups-based method, we propose a sequential random model interpolation method during training. We differ from model soups and SWA methods in that our approach explicitly maintains the connectivity of each averaged model, at the cost of increased GPU memory usage during the forward propagation process. We have provided a detailed comparison of the computational overhead and GPU memory costs of various baseline methods in the Appendix. This innovative approach streamlines the training process by eliminating the need for subsequent model selection steps within the **model pool** (*i.e.*, local models to be interpolated), which traditionally consumes a considerable amount of computational resources and time. Instead, our method integrates the model selection and interpolation phases seamlessly into the local training process. This integration ensures that the models selected for interpolation are inherently aligned within the same low-loss valley. In each training round, random interpolation weights are assigned to the models within the model pool. Forward and backward propagation are performed using the interpolated model, updating the weights of the currently active model (*i.e.*, the newly added model ) (corresponding to Algorithm 1 Line 7), while previously added model weights remain fixed.

### 3.3.2 DIVERSITY TERM.

The diversity term is proposed to address the *redundant model training* issue of the previous soups-based methods by encouraging low-loss region expansion. In particular, the diversity indicator assesses the variability among models within the model pool by summing the distances between pairs of models. Greater distances between models indicate a higher degree of diversity, which correlates with enhanced model quality. This diversity metric is integrated into the FL local training process as a regularization term to facilitate the extensive enlargement of low-loss regions, consequently maximizing the effectiveness of trained models. The diversity term (in Algorithm 1 Line 8) measures the distance between the current training model and other models that will be averaged, and we hope that this distance to be large. The diversity loss can be defined as

$$\ell_{\text{diversity}} = dist(f, \mathcal{M}) = \frac{1}{N} \sum_{n=1}^{N} dist(f, f_n). \tag{4}$$

Here, $f_n$ belongs to local interpolated model pool $\mathcal{M}$ and $N$ is the number of local candidate models. The candidate models (*i.e.*, model soups ingredients) are models to be interpolated in local training, and the model pool is the set of local candidate models (see Algorithm 1 Line 5). We use the $\ell_2$ norm to measure the distance between model weights.

### 3.3.3 AFFINITY TERM.

The affinity term is proposed to control the expansion of low-loss regions and prevent local candidate model training divergence. The affinity indicator assesses the level of alignment between each candidate model within the model pool and the initial global model by calculating the cumulative distances between each candidate model and the initialization model. Smaller distances between models signify a stronger affinity, indicating higher model quality. To ensure the controlled expansion of low-loss regions and reduce the probability of overlapping connected regions, this affinity metric is integrated into the training process as a regularization term. The affinity term (in Algorithm 1 Line 8) measures the distance between the candidate model and the initial model weights, with the aim of minimizing this dissimilarity (maximize this loss term) to ensure that the distance remains relatively small. The affinity loss can be defined as

$$\ell_{\text{affinity}} = dist(f, f_p). \tag{5}$$

Here, $f_p$ is a pre-trained model in the first communication round ($R = 1$). Moreover, it encourages each local model to lie in a close zone in the parameter space, which is beneficial for subsequent server aggregation, especially under data heterogeneity. We use $l_2$ distance for the $dist(,)$ metric for both Eq. 4 and Eq. 5.

### 3.3.4 OVERALL PIPELINE.

We outline *LSS* as follows: We begin with the initialization of the client's local model with the pretrained global model. Then we will refine the local model using affinity and diversity loss. This step is performed for a few local update steps. Finally, after updating local model, we aggregate them in the server following the common averaging operation in FedAvg (McMahan et al., 2017). The flow of *LSS* for local updating (Step 2 described in Sec 3.1) can be found in Algorithm 1.

In conclusion, our method aims to minimize the distance between the local fine-tuned model and the pre-trained initialized global model while maximizing the distance between the model soups ingredients (*i.e.*, the models to be averaged). Our fine-tuned models find large low-loss regions on their respective local datasets while ensuring parameters close to the pre-trained initialization. It is intuitive that the parameters of our fine-tuned models can be more easily aligned with those of models fine-tuned on similar datasets, thereby improving communication efficiency.

## 4 EXPERIMENT

### 4.1 EXPERIMENTAL SETUP

**Dataset.** Our experimental section considers two scenarios of Non-IID settings, namely label shift and feature shift. The label shift scenario investigates datasets such as FMNIST (Xiao et al., 2017)

Table 1: Label shift test accuracy after $R = 1$ and $R = 3$ communication rounds. We primarily compared two categories of methods: conventional FL methods and state-of-the-art local weight averaging-based fine-tuning methods that enhance domain generalization.

| | FMNIST | | CIFAR-10 | |
|---|---|---|---|---|
| Method | Accuracy ($R = 1$) ↑ | Accuracy ($R = 3$) ↑ | Accuracy ($R = 1$) ↑ | Accuracy ($R = 3$) ↑ |
| FedAvg 2017 | 35.54(1.71) | 90.04(0.32) | 58.34(0.86) | 66.74(0.76) |
| FedProx 2020 | 33.48(1.52) | 89.28(0.36) | 56.74(0.92) | 63.21(0.83) |
| MOON 2021a | 36.01(1.66) | 91.28(0.30) | 58.96(1.24) | 67.04(1.12) |
| FedBN 2021b | 34.20(1.73) | 89.87(0.47) | 57.04(0.75) | 64.51(0.67) |
| FedFomo 2021 | 33.94(1.65) | 88.41(0.69) | 55.01(0.89) | 62.69(0.75) |
| FedRep 2021 | 36.20(1.52) | 91.07(0.23) | 57.73(0.82) | 66.23(0.73) |
| FedBABU 2022 | 36.18(1.43) | 91.31(0.26) | 60.14(1.06) | 67.16(0.87) |
| SWA 2018 | 55.82(1.02) | 91.03(0.19) | 59.07(1.28) | 67.45(1.15) |
| SWAD 2021 | 58.66(0.87) | 91.22(0.16) | 60.54(1.15) | 67.65(0.97) |
| Soups 2022 | 60.11(0.64) | 91.56(0.24) | 61.00(1.04) | 67.63(0.94) |
| DiWA 2022b | 63.21(0.54) | 91.88(0.13) | 61.32(1.26) | 68.05(1.10) |
| *LSS* | **72.66**(0.73) | **92.45**(0.21) | **65.96**(1.50) | **75.16**(1.07) |

Table 2: Feature shift test accuracy after $R = 1$ and $R = 3$ communication rounds. *LSS* consistently outperforms other methods on both datasets across under feature shift settings.

| | Digit-5 | | DomainNet | |
|---|---|---|---|---|
| Method | Accuracy ($R = 1$) ↑ | Accuracy ($R = 3$) ↑ | Accuracy ($R = 1$) ↑ | Accuracy ($R = 3$) ↑ |
| FedAvg 2017 | 46.36(2.08) | 80.48(0.81) | 18.76(3.52) | 29.43(2.01) |
| FedProx 2020 | 44.01(1.92) | 77.83(0.68) | 17.27(3.22) | 27.18(2.29) |
| MOON 2021a | 50.11(1.72) | 83.02(0.64) | 19.61(3.54) | 31.27(2.34) |
| FedBN 2021b | 46.02(1.93) | 81.42(0.71) | 18.16(3.09) | 28.65(1.89) |
| FedFomo 2021 | 41.87(2.13) | 76.21(0.98) | 15.10(3.82) | 25.69(2.38) |
| FedRep 2021 | 47.43(1.73) | 82.02(0.63) | 18.89(2.60) | 30.42(1.84) |
| FedBABU 2022 | 48.02(1.81) | 83.20(0.79) | 19.44(2.43) | 32.06(1.88) |
| SWA 2018 | 54.13(0.72) | 85.33(0.62) | 22.07(2.55) | 35.90(1.61) |
| SWAD 2021 | 57.02(0.71) | 86.84(0.64) | 21.98(2.61) | 36.73(1.57) |
| Soups 2022 | 59.71(0.82) | 87.07(0.58) | 22.75(2.85) | 38.02(1.40) |
| DiWA 2022b | 61.54(0.83) | 88.83(0.69) | 24.88(2.54) | 38.32(1.50) |
| *LSS* | **72.86**(1.64) | **92.97**(0.65) | **27.86**(2.85) | **41.35**(1.46) |

and CIFAR10 (Krizhevsky et al., 2009), while feature shift involves Digit5 and DomainNet. Further information on the specific datasets can be found in the appendix. In the label shift scenario, we partitioned the dataset into five clients and the data for each client are sampled following Dirichlet distributions with coefficient $\alpha = 1.0$, yielding imbalanced label distributions. In the feature shift scenario, we utilized five clients for Digit5 (Ganin & Lempitsky, 2015; Li et al., 2021b) and five clients for DomainNet (Peng et al., 2019). Additional results on an extended number of clients are presented in the appendix.

**Model.** In terms of models, we used the ImageNet pre-trained ResNet50 (He et al., 2016) as the base model for the DomainNet dataset, while for other datasets, we used the pre-trained ResNet-18 trained on ImageNet (Deng et al., 2009). We also present the experimental results based on the vision transformer (ViT) model (Dosovitskiy et al., 2021) with parameter-efficient fine-tuning.

**Baselines.** We compare *LSS* against the vanilla FL method - FedAvg (McMahan et al., 2017) and several advanced FL algorithms designed for Non-IID settings, including FedProx (Li et al., 2020), MOON (Li et al., 2021a), FedBN (Li et al., 2021b), FedFomo (Zhang et al., 2021), FedRep (Collins et al., 2021) and FedBABU (Oh et al., 2022). Additionally, we make comparisons with top-performing weight/model-averaging-based domain generalization methods including SWA (Izmailov et al., 2018), SWAD (Cha et al., 2021), Soups (Wortsman et al., 2022) and DiWA (Ramé et al., 2022b) by adapting them to FL. In particular, the specific approach is to modify the local client training in the FedAvg framework to a corresponding fine-tuning approach. For more details, please refer to the appendix.

**Evaluation and implementation details.** Unless otherwise specified, the model performance in the experiments below refers to the global model performance after aggregation on the server side. Our training optimizer uses the Adam optimizer with a learning rate of $5e-4$ and a training batch size of $64$. For commonly used FL methods, due to the significant increase in local update steps that leads to worse convergence, we set their local update steps to $8$. For SWA, SWAD, and our method, we take more local update steps, with each model being averaged trained $8$ steps, and the default number of models to be averaged is $4$. For the Model Soups method and DiWA, we train 32 models with 8 steps. Additional details of experiment implementations are included in the Appendix.

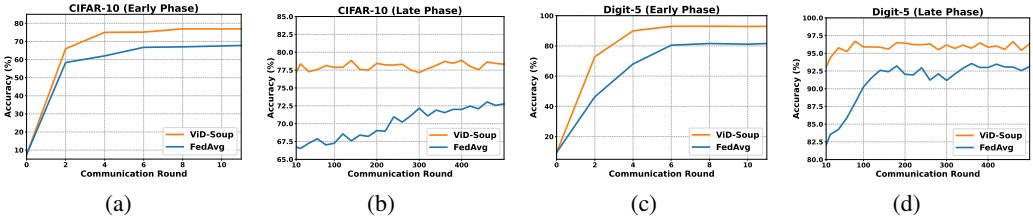

Figure 3: Convergence comparison of our proposed *LSS* with FedAvg. *LSS* achieves high accuracies much earlier (around 6 to 8 rounds) than FedAvg, which takes hundreds of communication rounds.

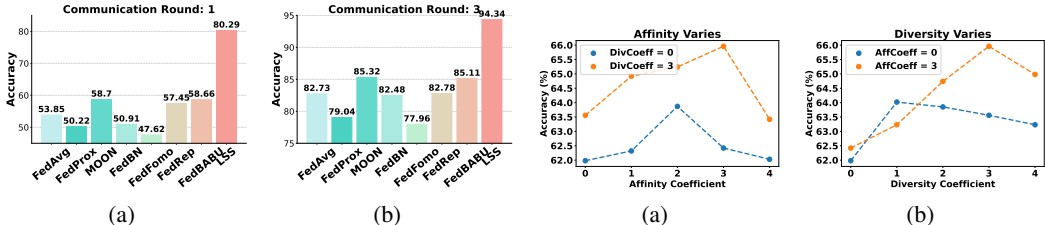

Figure 4: ViT + LoRA evaluation on Digit5.    Figure 5: Ablation on the affinity & diversity losses.

## 4.2 PERFORMANCE COMPARISON

**Results on label shift.** To demonstrate the effectiveness of *LSS* on label shift scenario, we conduct comparison experiments on FMNIST and CIFAR-10 datasets. We consider fine-tuning with an extremely limited number of communication rounds (*i.e.* $R = 1$ and $R = 3$). Table 1 reports the test accuracy with the format of mean (std) for all compared algorithms. All experiments are repeated 3 runs with different random seeds. In Table 1, *LSS* achieves the best accuracy on all settings of both datasets, which validates that *LSS* is efficient and effective in fine-tuning FL for label shift Non-IID. Notably, with just one round of communication, *LSS* can double the accuracy of the best Non-IID FL baseline method. Surprisingly, the simple extension of model-averaging-based domain generalization methods onto FedAvg (McMahan et al., 2017) (the 2nd big row in Table 1) perform very well, especially when the number communication round is small. The superior performance using local weight averaging-based fine-tuning is likely because it significantly reduces the gradient variance of local and global variance (see 3.2). We further provide results on different levels of label shift in the supplementary material.

**Results on feature shift.** Table 2 evaluates on feature shift scenario using Digits-5 and DomainNet datasets. Similar to the previous experiment setting for Table 1, we repeat all the algorithms with 3 random seeds. Consistent with the observation in Table 2, *LSS* is the top-performing method under all the settings for both datasets. We also observe better performance achieved by adapting model-averaging-based domain generalization methods (the 2nd big row in Table 2) in FL than the existing Non-IID FL methods (the 1st big row in Table 2), which further verifies the effectiveness of model averaging to obtain better global model while improving communication efficiency.

**Convergence plots.** We also evaluate the strength of *faster convergence* using the proposed *LSS* compared with FedAvg (McMahan et al., 2017) on CIFAR-10 (label shift) and Digtis-5 (feature shift). Fig. 3 depicts the testing accuracies at early and late phases regarding the number of communication rounds to reach convergence. First, by looking at the final testing accuracies on Fig. 3 (b) and (d), *LSS* achieves better performance. Second, Fig. 3 (a) and (c) show that *LSS* almost meets the targeted performance at the very early stage (*i.e.* around 6 to 8 rounds), whereas FedAvg requests over hundreds of communication rounds.

**Parameter-Efficient Tuning with ViT.** We also deployed the Vision Transformer (ViT) (Dosovitskiy et al., 2021) in FL learning. On Digits-5 dataset, we evaluate the ViT model with a resolution of 224 and a patch size of 16, which was pretrained on the ImageNet-21k dataset. Due to the large number of parameters in ViT, we used a parameter-efficient fine-tuning method called LoRA (Hu et al., 2022) to train it for all the methods. For more details about our ViT architecture and LoRA training, please refer to the appendix. It can be observed in Fig. 4 that our method is applicable to pre-trained ViT models, demonstrating that our approach can be combined with parameter-efficient fine-tuning methods to further enhance the communication efficiency of FL.

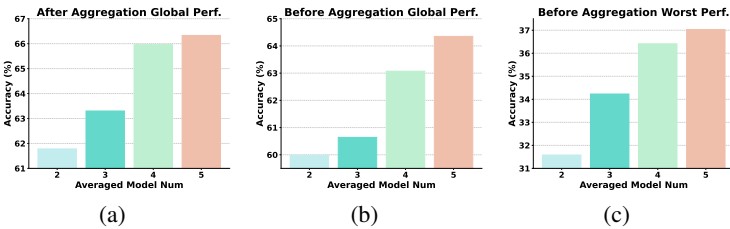

Figure 6: Ablation studies on the impact of the number of averaged models on communication efficiency and performance variance. We evaluated the influence of varied model quantities on global and averaged local model performance, as well as generalization on the worst client.

## 4.3 Ablation Studies

We conducted ablation experiments on the main components (i.e., affinity, diversity term and averaged model quantity) of our proposed method and evaluated their performance on the CIFAR dataset, with the performance metric being the global model performance at communication round $R = 1$.

**Investigation on regularization losses.** In order to examine the importance of affinity loss and diversity loss, as well as the influence of their corresponding coefficients, we adjust one coefficient within a range of 0 to 4 while maintaining the other at a constant value. By comparing the performance with and without loss term, we observe that adding affinity and diversity terms can enhance the model's performance. Additionally, we observe that the two terms complement each other, and selecting appropriate coefficients can achieve significant performance improvement (e.g., adjusting the affinity coefficient to 3 as shown in Fig. C.1 and diversity coefficient to 3 as shown in Fig. 5(b)).

**Investigation on the number of averaged models.** To investigate the impact of the averaged model quantity on enhancing communication efficiency and reducing gradient variance between local and global, we experiment with varied model quantities and evaluate their influence on global model performance, averaged local model performance[1], and worst out-of-distribution (OOD) generalization performance on the other clients. Fig. 6 shows that increasing the number of averaged models can improve the model's OOD generalization ability and enhance the performance of the aggregated model. This similar upward trend confirms the validity of our analysis linking OOD generalization and local-global variance. We provide a more detailed analysis on connecting our proposed *LSS* and OOD generalization in appendix C. Additionally, we can observe that increasing the number of models in our method can improve both pre-aggregation and post-aggregation model performance.

## 5 Conclusion

We propose an efficient method, Local Superior Soups (*LSS*), to reduce communication rounds in FL with pre-trained initialization, addressing the challenge of data heterogeneity. By employing sequential model interpolation, connectivity preservation, and two regularization terms (diversity and affinity), the method allows for an increase in local training steps and a reduction in communication rounds while avoiding client drift. This approach, tailored for pre-trained model adaptation in FL, offers training and inference efficiency, making it suitable for practical deployment in edge computing scenarios. As the first step towards understanding and developing model soups-based methods in pre-trained models in FL, this study conducts experiments on benchmark datasets. Our method attain superior performance with a only few rounds of communication and surpasses the performance of standard FL methods significantly across four datasets and under two distribution shift scenarios.

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

**Roadmap of Appendix** The appendix is organized as follows. We list the notations table in Section A. We provide the theoretical proof of the convergence analysis in Section B. We present the theoretical intuition of our proposed two loss term C. Next, we provide more detailed related work in Sec. D We present more experiment details and results in Sec. E.

## A NOTATION TABLE

Table 3: Important notations used in the paper.

| Notations | Description |
|-----------|-------------|
| $f$ | model parameters |
| $l$ | learning procedure |
| $m$ | feature dimension |
| $n$ | number of samples |
| $x$ | a sample |
| $y$ | a label |
| $\mathcal{D}$ | set of training domain |
| $\mathcal{L}$ | loss function |
| $M$ | number of clients |
| $N$ | number of averaged models |
| $R$ | total communication rounds |
| $X$ | input space of data |
| $Y$ | label space |
| $\lambda$ | coefficient for local training regularization term |
| $\tau$ | local training steps |

## B CONVERGENCE ANALYSIS

### B.1 FORMAL RESTATEMENT OF CONVERGENCE THEOREM

Standard FL (McMahan et al., 2017) employs a server to coordinate the following iterative distributed training:

*Step 1* In each global round of training $r \in [R]$, the server broadcasts its current global model weight $f_g^{(r-1)}$ to all the clients;

*Step 2* The selected client $c$ copies the current server model weight $f_c^{r,0} \leftarrow f_g$, performs $\tau$ local step updates, then sends $f_c^{r,L} - f_g^{(r-1)}$ back to the server;

*Step 3* The server aggregates the updates from all clients $\{f_c^{r,\tau} - f_g^{(r-1)}\}_{c=1}^C$ to form the new server model using the weighted averaging in Eq 1:

Note that the initialization $f^{(0,0)}$ ,with the subscription indicating model at $0$-$th$ communication round and $0$-$th$ local step, is a pre-trained model (*e.g.* using public datasets) in our problem. This work focus on improving *Step 2* to explore a larger low-loss region in local clients.

Formally, we present the convergence results (Theorem 1 in Wang et al. (2021a) and ours) and specify the following formal assumptions: 1) *Convexity and Smoothness Assumption* on $\beta$-smooth loss function, 2) *Bounded Variance of Stochastic Gradient Assumption* and 3) *Bounded Variance of Local and Global Gradient Assumption*).

**Assumption B.1.** *(Convexity and Smoothness).* $\mathcal{L}_i$ *is convex and $\beta$-smooth for all $i \in [M]$, i.e.,*

$$\|\nabla\mathcal{L}_i(w) - \nabla\mathcal{L}_i(v)\| \leq \beta\|w - v\|,$$

*for all $w, v$ in its domain and $i \in [M]$.*

**Assumption B.2.** *(Bounded variance of stochastic gradient). Each client can achieve an unbiased stochastic gradient with $\sigma^2$-uniformly bounded variance for all $k \in [0, \tau)$, namely*

$$\mathbb{E}[g_i(f_i^{(r,k)})|f_i^{(r,k)}] = \nabla\mathcal{L}_i(f_i^{(r,k)}), \quad \mathbb{E}[\|g_i(f_i^{(r,k)}) - \nabla\mathcal{L}_i(f_i^{(r,k)})\|^2|f_i^{(r,k)}] \leq \sigma^2. \quad (6)$$

**Assumption B.3.** *(Bounded variance of local and global gradient). The difference of local gradient* $\nabla \mathcal{L}_i(f)$ *and the global gradient* $\nabla \mathcal{L}(f)$ *is bounded in* $\ell_2$ *norm, namely*

$$\max_i \sup_f \left\| \nabla \mathcal{L}_i(f_i^{(r,k)}) - \nabla \mathcal{L}(f_i^{(r,k)}) \right\| \leq \zeta. \tag{7}$$

Following Wang et al. (2021a), we have the main theorem on converngence rate as follows. For the complete proof, please refer to Wang et al. (2021a). The main theorem on convergence rate as follows.

**Theorem B.1** (**Theorem 1**, Convergence Rate for Convex Local Functions Wang et al. (2021a))**.** *Under the aforementioned assumptions, we have*

$$\mathbb{E}\left[ \frac{1}{\tau R} \sum_{r=0}^{R-1} \sum_{k=1}^{\tau} \mathcal{L}(\overline{f}^{(r,k)}) - \mathcal{L}(f^\star) \right] \leq \frac{d^2}{2\eta\tau R} + \frac{\eta\sigma^2}{M} + 4\tau\eta^2\beta\sigma^2 + 18\tau^2\eta^2\beta\zeta^2. \tag{8}$$

*Further, when the client learning rate is chosen as*

$$\eta = \min\left\{ \frac{1}{4\beta}, \frac{M^{\frac{1}{2}}d}{\tau^{\frac{1}{2}}R^{\frac{1}{2}}\sigma}, \frac{d^{\frac{2}{3}}}{\tau^{\frac{2}{3}}R^{\frac{1}{3}}\beta^{\frac{1}{3}}\sigma^{\frac{2}{3}}}, \frac{d^{\frac{2}{3}}}{\tau R^{\frac{1}{3}}\beta^{\frac{1}{3}}\zeta^{\frac{2}{3}}} \right\}, \tag{9}$$

*we have*

$$\mathbb{E}\left[ \frac{1}{\tau R} \sum_{r=0}^{R-1} \sum_{k=1}^{\tau} \mathcal{L}(\overline{f}^{(r,k)}) - \mathcal{L}(f^\star) \right] \leq \frac{2\beta d^2}{\tau R} + \frac{2\sigma d}{\sqrt{M\tau R}} + \underbrace{\frac{5\beta^{\frac{1}{3}}\sigma^{\frac{2}{3}}d^{\frac{4}{3}}}{\tau^{\frac{1}{3}}R^{\frac{2}{3}}} + \frac{19\beta^{\frac{1}{3}}\zeta^{\frac{2}{3}}d^{\frac{4}{3}}}{R^{\frac{2}{3}}}}_{\text{errors from local updates \& Non--IID data}}. \tag{10}$$

Here, $d := \|f^{(0,0)} - f^\star\|$ refers to the distance between initialization and the global optimum $f^\star$.

## B.2 PROOF OF LEMMA 1

**Lemma 1.** Under the data heterogeneity setting, when the total number of gradient computations across all clients ($K = M\tau R$) is fixed and the local steps $\tau$ satisfies

$$\tau \leq \frac{\sigma}{\zeta} \sqrt{\frac{\sigma}{d\beta} \frac{K^{\frac{1}{2}}}{M^2}}, \tag{11}$$

the error upper bound Eq.equation 11 will be dominated by the second term $\mathcal{O}(1/\sqrt{K})$.

Taking local steps can save total communication rounds compared to synchronous SGD. To be more specific, as suggested in Wang et al. (2021a), when the total number of gradient evaluations/computations across all clients ($K = M\tau R$) is fixed and the local steps $\tau$ satisfies:

$$\tau \leq \min\left\{ \frac{\sigma}{d\beta} \frac{K^{\frac{1}{2}}}{M^2}, \frac{\sigma}{\zeta} \sqrt{\frac{\sigma}{d\beta} \frac{K^{\frac{1}{2}}}{M^2}} \right\}. \tag{12}$$

When the upper bound of local steps (Eq.(12)) becomes larger, there will be more communication savings. Therefore, the quantity in Eq.(12) represents the largest savings in communication rounds. Next, we show the error upper bound under the data heterogeneity setting.

*Proof.* Under high data heterogeneity, we have $\zeta \geq \sigma$, and:

$$1 \leq \frac{\sigma}{\zeta} \sqrt{\frac{\sigma}{d\beta} \frac{K^{\frac{1}{2}}}{M^2}} \leq \sqrt{\frac{\sigma}{d\beta} \frac{K^{\frac{1}{2}}}{M^2}} \leq \frac{\sigma}{d\beta} \frac{K^{\frac{1}{2}}}{M^2} \tag{13}$$

Therefore, we have Lemma 1:

$$\tau \leq \frac{\sigma}{\zeta} \sqrt{\frac{\sigma}{d\beta} \frac{K^{\frac{1}{2}}}{M^2}}, \tag{14}$$

$\square$

This Lemma 1 indicates that when client data are Non-IID, the side effects of the error term in the Theorem B.1 will be further exacerbated, therefore, increasing the local iteration steps effectively reduces the communication rounds.

**Why connected low-loss valley + pre-trained initialization can achieve extreme communication rounds reduction?** Based on the analysis above, we find that simply increasing the number of local training steps is insufficient for achieving extreme communication efficiency. The key lies in reducing the error term introduced by local updates. Importantly, to achieve a significant reduction in communication rounds, our primary focus should be on decreasing the last term of the RHS of Formula 2. This is because, under the extreme communication round reduction condition (*e.g.*, $R = 1$), the denominators of the first three terms all involve the local training steps $\tau$. As $\tau$ approaches infinity, the influence of these error terms can be eliminated, but the last error term remains. This term is mainly affected by three factors: local and global gradient dissimilarity $\zeta$, distance between initialized and optimal weights $d$, and the Lipschitz constant $\beta$ related with smoothness. Previous research (Nguyen et al., 2022) has shown that FL training based on pre-training initialization can better align updates from different clients, reducing the $\zeta$ term, which represents the difference between local and global gradients. Additionally, due to the characteristics of existing overparameterized models (Chizat et al., 2019; Li & Banerjee, 2021), the optimal solution is typically near the initialized point, leading to a very small $d$ term. As for the smoothness $\beta$ term, intuitively, if clients are trapped in isolated low-loss valleys, this situation reflects the non-smoothness of the local model function. By encouraging the regularization of training to find connected low-loss regions, we can effectively reduce the potential maximum value of the $\beta$ term during the training process. Through the above analysis, we conclude that pre-training initialization combined with our regularization training that encourages the search for connected regions can reduce the error terms introduced by local updates, thus increasing the upper limit of local training steps and achieving the goal of reducing communication rounds.

## C  THEORETICAL INTUITIONS.

### C.1  DECOMPOSITION OF GENERALIZATION BOUND

**Connecting $\zeta$ with out-of-distribution error.** ensemble is a category of the promising method that ensembles trained models to improve generalizability as demonstrated in centralized settings via reducing model discrepancy (Izmailov et al., 2018). To reduce the variance $\zeta$ of local and global gradients that is resulted by data heterogeneity, we aim to adapt ensemble to FL. Intuitively, local client training that can reduce the error on the worst domain (client) in FL will reduce the variance $\zeta$.

In the following, we detail how to reduce $\zeta$ with OOD error with a bias-variance-covariance-locality (BVCL) decomposition analysis. ensemble can be defined as: $f_{\text{WA}} \triangleq 1/N \sum_{n=1}^{N} f_n$. We have the following decomposition of ensemble's expected test error. *Bias-variance-covariance-locality decomposition.* The expected generalization error on domain $T$ of $f_{\text{WA}}$ over the joint distribution $(L_S^N \triangleq \{l_S^{(N)}\}_{N=1}^N)$ of $N$ learning procedure on domain $S$ is:

$$\mathbb{E}_{L_S^N} \mathcal{E}_T(f_{\text{WA}}(L_S^N)) = \mathbb{E}_{(x,y)\sim p_T}\left[\text{bias}^2(x,y) + \frac{1}{N}\text{var}(x) + \frac{N-1}{N}\text{cov}(x)\right] + O(\bar{\Delta}^2), \quad (15)$$

Here, cov refers to the covariance of predictions made by two member models. The first component is the same bias as that of each individual member. The variance of ensemble is split into two parts: the variance of each member divided by the number of members ($N$) and a covariance term. The last locality term enforces constraints on the weights to ensure the functional ensembling approximation remains valid. In summary, combining $N$ models reduces variance by a factor of $N$, but introduces the covariance and locality terms which must be controlled to ensure low OOD error.

In the analysis presented in Ramé et al. (2022b), the authors proposed a BVCL decomposition based on the approximation of functional ensembling (i.e., averaged prediction instead of parameter) by WA. The expected generalization error on domain $T$ of $f_{\text{WA}}$ over the joint distribution $(L_S^N \triangleq \{l_S^{(N)}\}_{N=1}^N)$ of $N$ learning procedure on domain $S$ is:

$$\mathbb{E}_{L_S^N} \mathcal{E}_T(f_{\text{WA}}(L_S^N)) = \mathbb{E}_{(x,y)\sim p_T}\left[\text{bias}^2(x,y) + \frac{1}{N}\text{var}(x) + \frac{N-1}{N}\text{cov}(x)\right] + O(\bar{\Delta}^2), \quad \text{(BVCL)}$$

**Definition C.1** (Bias). *For $x \in X$ and $y \in Y$, we define the bias of OOD prediction as,*

$$\mathrm{bias}(x, y) = y - \mathbb{E}_{l_S}[f(x, l_S)]. \tag{16}$$

**Definition C.2** (Variance). *For $x \in X$, we define the variance of prediction as*

$$\mathrm{var}(x) = \mathbb{E}_{f_S}\left[ (f(x, l_S) - \mathbb{E}_{l_S}[f(x, l_S)])^2 \right]. \tag{17}$$

**Definition C.3** (Covariance). *For $x \in X$, we define the covariance of prediction produced by two different learning procedures $l_S$ and $l'_S$ as*

$$\mathrm{cov}(x) = \mathbb{E}_{l_S, l'_S}\left[ (f(x, l_S) - \mathbb{E}_{l_S}[f(x, l_S)])(f(x, l'_S) - \mathbb{E}_{l_S}[f(x, l_S)]) \right]. \tag{18}$$

**Definition C.4** (Locality). *For any averaged models $f_i$ (for $i \in [N]$), $i$ is the index of an averaged model, $N$ is the total number of averaged models, we define the locality of all averaged models as*

$$\bar{\Delta}^2 = \mathbb{E}_{L_S^N} \Delta_{L_S^N}^2 \ \text{with} \ \Delta_{L_S^N} = \max_{i=1}^{N} \|f_i - f_{WA}\|_2. \tag{19}$$

Following the definitions of the terms in the BCVL generalization bound, we discuss the insights of reducing the bound via the proposed strategy. Our method is based on WAFT, which enjoys the benefit of reducing prediction variance by averaging the predictions of multiple models. The diversity term in our proposed method reduces the covariance term by encouraging functional diversity in the parameter space. The affinity term in our proposed method reduces the locality term to ensure the approximation of weight averaging (WA) to prediction ensembling.

**Analysis on variance.** One can see that an increase in the number of averaged models can directly lead to a reduction in variance. The straightforward averaging $M$ models, as seen in the vanilla WAFT method, diminishes variance by a factor of $M$. However, this approach also introduces covariance and locality terms, which necessitate meticulous management on adding new averaged models to guarantee minimal out-of-distribution (OOD) error.

**Analysis on covariance.** The covariance term represents the predictive covariance between two member models whose weights are averaged. It increases when the predictions of different averaged models are highly correlated. In the worst-case scenario where all predictions are identical, the covariance is equal to the variance, rendering the benefits of weight averaging ineffective Ramé et al. (2022b). Conversely, when the covariance is lower, the advantages of weight averaging over individual models become more pronounced. Therefore, it is crucial to address covariance by promoting functional diversity among the averaged models. Our proposed method incorporates a diversity term that aims to reduce this covariance.

**Analysis on locality.** The locality term, which represents the expected squared maximum distance between weights and their average, constrains the weights to be close and ensures the approximation. The affinity term in our proposed method encourages the reduction of this locality term.

Overall, to reduce WA's error in OOD, we need to seek a good trade-off between diversity and locality. Our solution achieves this balance through two optimizable loss terms, the diversity term, and the affinity term. Besides, the direct combination of $M$ models, as in the vanilla WAFT method, reduces variance by a factor of $M$ but introduces covariance and locality terms that need to be carefully managed in order to ensure low OOD error.

It is worth noting that, from an implementation perspective, unlike the model soups method (see Fig. 7 middle), which requires retraining a large number of candidate models for model selection and interpolation, our method only selects a few models (typically 3 to 5) for sequential random interpolation training in order to maintain connectivity. This significantly reduces the time cost of local training. Furthermore, unlike model ensembles (see Fig. 7) that require storing multiple model weights and integrating predictions during inference, our method only needs to retain an averaged weight during the final inference stage. This greatly reduces the memory footprint and enhances the inference speed on the client side.

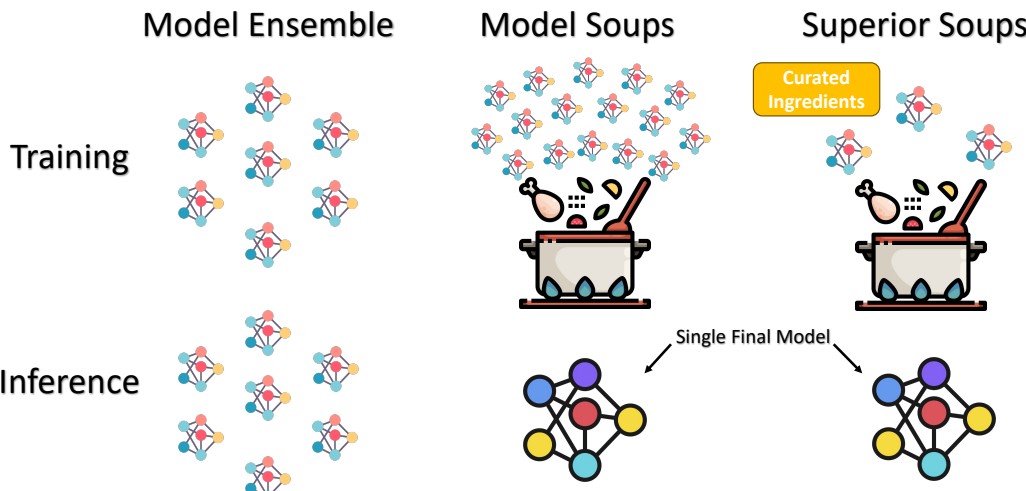

Figure 7: Comparison on model ensemble, model soups, and superior soups.

# D   MORE RELATED WORK

## D.1   HETEROGENEOUS FEDERATED LEARNING

FL performance downgrading on Non-IID data is a critical challenge. A variety of FL algorithms have been proposed to handle this heterogeneous issue. From an optimization perspective: FedProx (Li et al., 2020) adds $L_2$ norm to the client model and the previous server model to regularize them. This helps to prevent the client models from diverging too far from the server model. Scaffold (Karimireddy et al., 2020) adds a variance reduction term to mitigate the "clients-drift." MOON (Li et al., 2021a) uses mode-level contrastive learning to stabilize local training by making the client models more robust to changes in the data distribution. In addition, personalized FL (Tan et al., 2021) is another approach to achieving high local testing performance on Non-IID data. For aggregation perspective: FedBN (Li et al., 2021b) uses local batch normalization to alleviate the feature shift before averaging models. For extreme communication efficient: In recent years, there have been some FL methods based on one-shot communication rounds. These methods typically use additional techniques on the server-side, such as using prediction ensembles (Guha et al., 2019) instead of weight ensembles or generating data (Zhang et al., 2022a; Heinbaugh et al., 2023) from local models for centralized training, to improve the performance of the aggregated model. These methods are orthogonal to our client training-based approach. There are also works on few-round communication rounds in FL based on meta-learning frameworks (Park et al., 2021), but the data partition used in the experimental setup may not be suitable for practical FL scenarios.

## D.2   FINE-TUNING AND MODEL INTERPOLATION

Fine-tuning aims to achieve improved performance on the given task by leveraging the learned knowledge of the pre-trained model. Choshen et al. (2022) empirically study the impact of fine-tuning from a pre-trained model in FL and unsurprisingly find that starting from a pre-trained model reduces the training time required to reach a target error rate and enables the training of more accurate models than starting from random initialization. Zhang et al. (2022b) propose a knowledge distillation approach for fine-tuning the global model, called FedFTG. In addition, fine-tuning in FL has been widely used in personalized FL to address Non-IID problems by having each user adapt the global model to personalized local models using their own data. For example, FedBABU (Oh et al., 2022) splits the model into body and head, then fine-tuning the head part for personalization. Cheng et al. (2021) propose FTFA and RTFA that start with a pre-trained model and then fine-tunes a small subset of model parameters using the FedAvg (McMahan et al., 2017) algorithm. However, this line of work focuses on optimizing local performance and ignores the generalization of global data. This can lead to a performance drop when we further update the global model from the updated local models. Weight averaging and model recycling are not only efficient ways to aggregate machine

learning models but also present promising benefits of improving model generalizability. Inspired by the linear mode connectivity property of neural networks trained with stochastic gradient descent (SGD) (Nagarajan & Kolter, 2019; Frankle et al., 2020), Model Soups (Wortsman et al., 2022) proposes to combine many independent runs with varied hyper-parameter configurations. Similarly, DiWA (Ramé et al., 2022b) utilizes this idea of Model Soups while theoretically analyzing the importance of training different models with diverse hyper-parameters within mild ranges. Soups-based methods (Wortsman et al., 2022; Ramé et al., 2022b) rely on aggregating diverse models to improve model generalizability. To induce greater diversity, some methods such as (Maddox et al., 2019) using a high constant learning rate, (Wortsman et al., 2021) minimizing cosine similarity between weights, (Izmailov et al., 2019) using a tempered posterior and model Ratatouille (Ramé et al., 2022a) averages diverse model trained from auxiliary datasets.

# E EXPERIMENT DETAILS

## E.1 EXPERIMENTAL SETUP DETAILS

**Dataset.** We validate the effectiveness of our proposed method with four datasets, FMNIST Xiao et al. (2017), CIFAR-10 Krizhevsky et al. (2009), Digit-5 Ganin & Lempitsky (2015); Li et al. (2021b), and DomainNet Peng et al. (2019). The Fashion-MNIST (FMNIST) dataset is a dataset of Zalando's article images consisting of a training set of $60,000$ examples and a test set of $10,000$ examples. Each example is a $28 \times 28$ grayscale image of a piece of clothing. The dataset is divided into $10$ classes: t-shirt/top, trouser, pullover, dress, coat, sandal, shirt, sneaker, bag, and ankle boot. The CIFAR-10 dataset is a popular dataset for machine learning research. It consists of $60,000$ $32 \times 32$ color images divided into $10$ classes: airplane, automobile, bird, cat, deer, dog, frog, horse, ship, and truck. The dataset is split into $50,000$ training images and $10,000$ test images. The Digit-5 dataset is a collection of five popular digit datasets, MNIST Deng (2012) (55000 samples), MNIST-M (55000 samples), Synthetic Digits Ganin & Lempitsky (2015) (25000 samples), SVHN (73257 samples), and USPS (7438 samples). Each digit dataset includes a different style of 0-9 digit images. The DomainNet dataset is a large-scale dataset of images collected from six different domains: clipart, infograph, painting, quickdraw, real, and sketch. The dataset contains $600,000$ images, each labeled with one of $345$ object categories. The images in the DomainNet dataset are of high quality and are diverse in terms of their content and style.

**Model.** We used the pre-trained models from the timm repo [1], which are a collection of state-of-the-art deep learning models for computer vision tasks. For our proposed *LSS*, we use Adam optimizer with a learning rate of $5e-4$, momentum $0.9$, and weight decay $5e-4$. The default number of averaged models is $4$. Each model updates $8$ epoch then aggregates with the others. The default affinity term coefficient is $3$ and diversity term coefficient is $3$. We set the batch size to $64$ by default. For vision transformer (ViT) Dosovitskiy et al. (2021) model, we adopt ViT base model with $224 \times 224$ image size and $16 \times 16$ input patch size. The ViT is a neural network architecture for image classification that uses a self-attention mechanism to learn the relationships between pixels in an image. ViT has been shown to achieve state-of-the-art results on a variety of image classification benchmarks, including ImageNet and CIFAR-10.

**Training Details.** We implement all the methods in PyTorch, and we run all the experiments on an NVIDIA Tesla V100 GPU. Unless otherwise specified, the model performance in the experiments below refers to the global model performance after aggregation on the server side. For commonly used FL methods, due to the significant increase in local update steps that leads to worse convergence, we set their local update steps to $8$.

**Applying WAFT to FL Local Update.** For SWA Izmailov et al. (2018), SWAD Cha et al. (2021), and our method *LSS*, we take more local update steps, with each model being averaged trained $8$ steps, and the default number of models to be averaged is $4$. For the Model Soups Wortsman et al. (2022) method and DiWA Ramé et al. (2022b), we trained 32 models and each model trained 8 steps. The hyper-parameter configuration for model selection includes learning rate ($[1e-4, 5e-4, 1e-5]$), batch size ($[32, 64, 128]$), dropout rate ($[0.0, 0.1, 0.3]$), and weight decay $[5e-4, 5e-5, 5e-6]$. Each run randomly select one of the hyper-parameter options. From each run of WAFT method, we take

---

[1]https://github.com/huggingface/pytorch-image-models

the weights of the epoch with maximum accuracy on the validation dataset, which follows the training distribution.

## E.2 EXTENDED EXPERIMENT RESULTS

**Arbitrarily increasing local steps cannot reduce communication rounds.**

From Table 4, we can see that simply increasing local steps does not always lead to improved model performance. For FedAvg on the CIFAR10 dataset, increasing local steps beyond 8 actually results in a decrease in model performance.

Table 4: FedAvg with different local steps: Label shift test accuracy after $R = 1$ communication rounds (CIFAR-10 with 5 Clients).

| Method | Accuracy ($\tau = 1$)↑ | Accuracy ($\tau = 4$)↑ | Accuracy ($\tau = 8$)↑ | Accuracy ($\tau = 12$)↑ | Accuracy ($\tau = 16$)↑ |
|---|---|---|---|---|---|
| FedAvg 2017 | 34.03(2.84) | 49.08(1.51) | **58.34**(0.86) | 55.76(0.82) | 53.21(0.80) |

**Computational and memory costs comparison.**

In Table 5, we provide detailed information on computational overhead and memory usage for various methods. Since the computational overhead and memory usage of FedAvg and other used FL methods are nearly identical, we only present the data for FedAvg here. Similarly, as the computational overhead and memory usage for SWA and SWAD, as well as for Soups and DiWA, are also nearly the same, we only show the data for SWA and Soups methods. It can be observed that our method requires more memory compared to other soups-based methods. However, the overall computational time for a single client's communication round is faster in our approach. This is because other soups-based methods require training a large number of models repeatedly to achieve good model performance. For instance, Soups needs to train 32 models, whereas our method only requires training 4 models. If the number of models trained by Soups is reduced to just 4, it only brings about a 5% improvement compared to FedAvg with a communication round of 1.

Table 5: Computational and memory costs of different types of method (ResNet-18).

| Costs | FedAvg 2017 | SWA 2018 | Soups 2022 | *LSS* ($M = 2$) | *LSS* ($M = 4$) |
|---|---|---|---|---|---|
| MACs (G) | 1.82 | 1.82 | 1.82 | 2.73 | 4.55 |
| Train Time Per Epoch (s) | 2.66 | 2.73 | 2.66 | 12.27 | 20.43 |
| Train Time Per Round (s) | 21.28 | 433.31 | 683.52 | 100.98 | 169.77 |

*LSS* **encourages smoothness (reducing** $\beta$**).** In Table 6, we provide the performance degradation of trained models evaluating under varying levels of random noise. Generally, a smaller performance degradation indicates a more robust model, which to some extent reflects the smoothness of the trained model. We can observe that our method exhibits greater robustness to noise perturbation.

Table 6: Smoothness of the trained model. Evaluated trained model performance drop on a testset with added $\ell_0$ norm random noise. CIFAR-10 dataset Dirichlet distribution $\alpha = 1.0$ and $\alpha = 0.1$: Label shift test accuracy after $R = 1$

| | **CIFAR-10** ($4/255$) | | **CIFAR-10** ($8/255$) | |
|---|---|---|---|---|
| Method | Accuracy ($R = 1$)↓ | Accuracy ($R = 3$)↓ | Accuracy ($R = 1$)↓ | Accuracy ($R = 3$)↓ |
| FedAvg 2017 | 1.30 | 1.17 | 3.06 | 2.93 |
| *LSS* | 0.89 | 0.76 | 2.37 | 1.85 |

*LSS* **improves flatness of loss landscape.** The sharpness measure utilized in the Table 7 computes the median of the dominant Hessian eigenvalue across all training set batches through the Power Iteration algorithm (Yao et al., 2020). This metric signifies the maximum curvature of the loss landscape, commonly employed in the literature on flat minima (Kaddour et al., 2022) to indicate sharpness. As

demonstrated in the presented table, it is clear that our proposed method results in flatter minima compared to FedAvg.

Table 7: Loss landscape flatness quantification with Hessian eigenvalue.

|  | FedAvg $\downarrow$ | *LSS* ($M = 2$) $\downarrow$ | *LSS* ($M = 3$) $\downarrow$ | *LSS* ($M = 4$) $\downarrow$ |
|---|---|---|---|---|
| Hessian Eigenvalue | 193.18 | 147.20 | 136.67 | **119.14** |

**Evaluation with more clients.** To assess the effectiveness of our method in larger-scale client scenarios, we conducted an expanded experiment involving 50 clients. From the Table 8, we can observe that our proposed method maintains a significant advantage across different client scales, particularly when the number of communication rounds is small ($R = 1$).

Table 8: Different client numbers (5 Clients and 50 Clients): Label shift test accuracy after $R = 1$ and $R = 3$ communication rounds.

|  | **CIFAR-10 (5 Clients)** | | **CIFAR-10 (50 Clients)** | |
|---|---|---|---|---|
| Method | Accuracy ($R = 1$) $\uparrow$ | Accuracy ($R = 3$) $\uparrow$ | Accuracy ($R = 1$) $\uparrow$ | Accuracy ($R = 3$) $\uparrow$ |
| FedAvg 2017 | 58.34(0.86) | 66.74(0.76) | 49.32(0.93) | 68.39(0.61) |
| *LSS* | **65.96**(1.50) | **75.16**(1.07) | **56.72**(0.53) | **73.32**(0.46) |

**Evaluation with ViT.** To validate the effectiveness of our method across different network architectures, we conducted an expanded experiment using the Vision Transformer (ViT) model based on the Transformer architecture. Upon observing the Table 9, it is evident that our method consistently enhances the communication efficiency of federated learning with ViT model architectures.

Table 9: Different Network Architecture (ResNet-18 and ViT): Label shift test accuracy after $R = 1$ and $R = 3$ communication rounds.

|  | **CIFAR-10 (ResNet-18)** | | **CIFAR-10 (ViT Base)** | |
|---|---|---|---|---|
| Method | Accuracy ($R = 1$) $\uparrow$ | Accuracy ($R = 3$) $\uparrow$ | Accuracy ($R = 1$) $\uparrow$ | Accuracy ($R = 3$) $\uparrow$ |
| FedAvg 2017 | 58.34(0.86) | 66.74(0.76) | 60.35(0.82) | 69.38(0.51) |
| *LSS* | **65.96**(1.50) | **75.16**(1.07) | **67.48**(0.70) | **76.81**(0.47) |

**Evaluation with different Non-IID level.** To further comprehensively validate the effectiveness of our method under different levels of data heterogeneity, we conducted experiments on the CIFAR-10 dataset by adjusting the coefficients $\alpha$ of the Dirichlet distribution. We examined the performance of our method in scenarios with greater distribution variations. Based on the Table 10, it is evident that our method maintains a significant advantage in scenarios with larger data heterogeneity.

Table 10: Different Non-IID level (Dirichlet distribution $\alpha = 1.0$ and $\alpha = 0.1$): Label shift test accuracy after $R = 1$ and $R = 3$ communication rounds.

| Method | **CIFAR-10** ($\alpha = 1.0$) | | **CIFAR-10** ($\alpha = 0.1$) | |
|---|---|---|---|---|
| | Accuracy ($R = 1$) $\uparrow$ | Accuracy ($R = 3$) $\uparrow$ | Accuracy ($R = 1$) $\uparrow$ | Accuracy ($R = 3$) $\uparrow$ |
| FedAvg 2017 | 58.34(0.86) | 66.74(0.76) | 18.30(2.25) | 45.85(1.24) |
| *LSS* | **65.96**(1.50) | **75.16**(1.07) | **26.70**(1.62) | **50.02**(0.82) |

