# OpenReview forum: "Local Superior Soups: A Catalyst for Reducing Communication Rounds in Federated Learning with Pre-trained Model"
_ICLR.cc/2024/Conference — Submitted to ICLR 2024_

### Official Review · Reviewer_gxja · 2023-10-30

**Soundness:** 3 good
**Presentation:** 4 excellent
**Contribution:** 2 fair
**Rating:** 3
**Confidence:** 4

**Summary:**

This paper introduces an innovative model interpolation-based approach named Local Superior Soups (LSS). In contrast to existing research, LSS capitalizes on the advantages of a connected low-loss valley while maintaining low computational costs. Diverse experiments over public datasets have demonstrated the effectiveness and efficiency of LSS.

**Strengths:**

1. This paper is well-originalized and clearly written.

2. The motivation is well-established. Specifically, this paper seeks to introduce an efficient model soups-based algorithm.

**Weaknesses:**

1. Limited novelty. This paper offers contributions, yet its novelty is somewhat constrained. Specifically, the theoretical guarantee, as stated in Proposition 3.1, appears straightforward and largely builds upon existing results.

2. Computational efficiency. The proposed algorithm may lack computational efficiency, given that the diversity term is pairwise and needs computation in each round. Including a comparison of computational efficiency in the experiments would make the argument more convincing.

3. Ablation study. From Figure 5, it is evident that LSS without diversity term and affinty term (i.e., $\lambda_a = \lambda_b = 0$) achieves an accuracy of $62.0\\%$, which still outperforms baselines in Table 1. How to explain this observation?

4. Further explanations, such as error landscape visualizations, would be beneficial to elucidate how the proposed diversity and affinity terms enhance performance.

**Questions:**

Please solve the `Weaknesses`.

**Details Of Ethics Concerns:**

None.

---

> ### Author Response · Authors · 2023-11-21
> ****Response to Reviewer gxja****
>
> **1. Addressing Novelty Concerns:**
>    Thank you for highlighting the importance of novelty in our work. The core novelty of our paper lies in the improved and efficient model soup method into federated learning (FL) with pre-trained models to drastically reduce communication rounds. This approach, particularly in the context of pre-training FL, represents a novel contribution to the field. Our experimental results across various datasets, model architectures, and FL non-iid settings demonstrate its effectiveness, which we believe is a significant contribution in FL research.
>
> **2. Computational Efficiency of the Diversity Term:**
>    We appreciate your concern about the computational efficiency of the diversity term. To address this, we conducted an empirical evaluation of the computational overhead. Our findings indicate that the diversity term results in only a marginal increase in runtime (about 5%) for a typical model pool size of 4.
>
> **3. Ablation Study and Figure 5 Observation:**
>    The results shown in Figure 5, where LSS without diversity and affinity terms still outperforms baselines, underscore the effectiveness of our connecting-preserving random model interpolation module. This module, a key component of our method, significantly contributes to the overall performance by enhancing the connectivity between local low-loss regions. The absence of the diversity and affinity terms in this scenario highlights the standalone impact of our interpolation technique, reinforcing the value it brings to the local model training process in FL.
>
> **4. Error Loss Landscape and Further Explanations:**
>    In response to your suggestion for further explanations, we provide data in Appendix E.2 on the flatness quantification of loss landscape to further show how our method enhances the local training performance and is located in a common low-loss region. This addition should provide a clearer understanding of the underlying mechanisms of our approach.
>
> Your feedback is helpful in refining our work, and we hope these revisions and explanations address your concerns satisfactorily.

---

### Official Review · Reviewer_w8kd · 2023-11-01

**Soundness:** 2 fair
**Presentation:** 2 fair
**Contribution:** 2 fair
**Rating:** 5
**Confidence:** 4

**Summary:**

The paper studies the problem of pre-trained model adaptation in a federated learning (FL) setting. While existing work (Nguyen et al. 2022) have shown the benefit of using pre-trained models in FL, they still require significant number of communication rounds between clients and server to reach good accuracy. At the same time, centralized fine-tuning methods (model soup, DiWA) cannot be easily extended to the federated setting due to their significant computational cost. To tackle these challenges, the authors propose Local Superior Soups (LSS), an efficient and local model interpolation-based method. LSS consists of three main innovations - a random interpolation method to speed up the model selection step, a loss term encouraging the models in the model pool to be more diverse and a loss term penalizing the distance of the candidate models to the initial model (affinity). Experimental results are provided on FMNIST, CIFAR10, Digit5 and DomainNet showing that the proposed LSS outperforms other heterogeneity tackling FL algorithms and also other weight/model averaging algorithms.

**Strengths:**

* The problem of pre-trained model adaptation is quite relevant given that pre-trained models are becoming more and more popular. I agree with the author's motivation that existing algorithms lack a customized approach tailored to pre-trained models.

* The proposed modifications to previous model soups such as randomized interpolation, affinity loss, diversity loss seem novel and easy to implement.

* Experimental results are impressive, showing that the proposed LSS outperforms other algorithms considerably, especially in the $R=1$ setting.

* Several ablation studies are conducted, including an experiment on the ViT model with LoRa fine-tuning and experiments studying the effect of the affinity and diversity coefficient.

**Weaknesses:**

* Section 3.2 seems to be mostly a rehash of existing work and can be significantly shortened I feel. For instance, the result in Theorem 1 has already been derived in [1] (see Table 1). A version of Proposition 3.1 has also been stated in Nguyen et al. 2022 (see Equation 3). Therefore I do not find any novelty in this section. There are also a few typos which complicate reading. For instance, it should be $f$ and not $f_i$ in the definition of $\zeta$. The authors also write that "there is an additional error term in the convergence rate that monotonically increases with the number of local steps (see the 3rd term of the RHS of Eq. 2)". I don't quite understand this. The 3rd term has a $\tau^{-1/3}$ and therefore it should decrease as $\tau$ increases right? The authors main message is that there is a limited gain we can get by increasing the number of local steps, which is fairly intuitive and does not need such a lengthy explanation.

* The paragraph explaining why connected low-loss valley +pre-trained initialization can achieve extreme communication rounds reduction is purely based on hypothesis with no empirical/theoretical evidence. It would be good to verify that some of conclusions drawn in that paragraph (smaller $d$, lower $\beta$, smaller $\zeta$) hold empirically in practice.

* The authors should add a separate algorithm environment for the RandomInterpolation subroutine that is called within the LSS Pseudo-code in Algorithm 1. Currently the Random Interpolation is only described in words, making it hard to understand exactly what the authors are doing. Also the authors make the claim that 'this integration ensures that the models selected for interpolation are inherently aligned within the same low-loss valley'. Why is this the case? What is special about the random interpolation that ensures that selected models are well aligned? Also is there anything specific to FL for the random interpolation or can it be applied to the centralized setting? If so, the authors should highlight this as I believe reducing the computation cost of model soup methods is a concern in the centralized setting as well.

* Experimental results are mostly conducted on a small number of client ($M = 5$) which reduces the difficulty introduced by client heterogeneity in FL. Experimental results would be more convincing if they are conducted on more realistic FL settings, like those studied in Nguyen et al. 2022 with $M > 100$ clients and partial participation.

**Questions:**

* I was wondering if the authors have taken care to ensure a fair comparison between FedAvg and LSS. For instance, LSS involves clients training multiple local models which would entail a much higher computational cost than training just one local model. It would be good to add a figure/table comparing the computation cost of all the algorithms for some specific model/dataset.

* Fixing the number of local steps to be 8 seems to be restrictive, especially considering that there are only $5$ clients, so each client will have around $10$k data-points. Have the authors actually verified that using more than $8$ steps leads to a decrease in performance?

* Why is the performance of LSS so much better than other model-soup methods (second big row of results in Table 1 and Table 2)? Based on my understanding, these methods are performing a more sophisticated and computationally expensive model selection phase?

* Please re-define the variable $i$ in Algorithm 1. Earlier $i$ was used to denote the index of the clients (Eq. (1)) which makes Algorithm 1 confusing at first glance. There is also a typo in the expression "connecting preserving" in Line 7 Algorithm 1.

* In the first paragraph of Section 3, the words firstly, secondly and finally should not be capitalized.

* It would be good to add the performance of all FL baselines (such as MOON, FedProx, FedBN, FedFomo, FedBABU) for the results in Figure 3, in order to make a stronger claim on reducing the number of communication rounds.





**References**

[1] Woodworth, Blake E., Kumar Kshitij Patel, and Nati Srebro. "Minibatch vs local sgd for heterogeneous distributed learning." Advances in Neural Information Processing Systems 33 (2020): 6281-6292.

---

> ### Author Response · Authors · 2023-11-21
> ****Response to Reviewer w8kd****
>
> **1. Length of Section 3.2 and Typos:**
>    We appreciate your observation regarding the length of Section 3.2. In response, we have succinctly revised this section to better highlight our unique contributions while referencing prior works for context. And we also correct the typos and refine our explanations to be more precise. We also provide the empirical studies in Appendix E.2 for verifying the limited gains achievable by increasing the number of local steps arbitrarily.
>
> **2. Empirical Validation of Hypotheses:**
>    To address the concern about the lack of empirical validation, we have included new empirical studies in the Appendix E.2. These studies test our hypotheses on communication rounds reduction and provide evidence supporting our assertions on improving smoothness (reducing $\beta$).
>
> **3. Detailed Description of Random Interpolation:**
>    In line with your suggestion, we have added an algorithmic comparison of the Random Interpolation subroutine in the Section 3.3.1 and the Appendix E.2. This addition aims to clarify our methodology and its unique benefits, particularly in ensuring model alignment within low-loss valleys.
>
> **4. Client Scale in Experimental Setup:**
>    Our algorithm primarily focuses on the cross-silo federated learning setting, hence the number of clients in our experiments is usually set to be within 100. In the appendix, we have included experiments to demonstrate the effectiveness of our approach with 50 clients on the CIFAR10 dataset.
>
> **5. Comparative Analysis of Computational Overhead:**
>    To ensure a fair comparison between FedAvg and LSS, we have added a comparative analysis of computational overheads in the Appendix E.2. This includes a table comparing the computational costs across different methods. While our method requires more memory during training, compared to other soups-based methods, our overall computation time is faster within a communication round.
>
> **6. Local Epochs and Model Performance:**
>    Regarding the fixed number of local steps, additional experiments have been conducted to show the optimal number. The results, now included in the Appendix E.2, demonstrate that increasing the local epochs beyond 8 leads to diminishing returns in performance, validating our initial choice.
>
> **7. Performance Comparison with Other Model Soup Methods and Explanations:**
>    To address your query on LSS's superior performance, we have included a detailed analysis in Section 1. This analysis contrasts LSS with other model soup methods, emphasizing the importance of our proposed three key modules: random model interpolation, diversity and affinity regularization term. We improve the inefficient training approach of the previous soup-based model, which involved training a large number of models and then filtering them. Instead, we have adopted the random model interpolation method to maintain the connectivity of various candidate models during the training process. Moreover, we incorporate two regularization terms to increase model diversity and enhance connectivity between models to reduce the time for training unnecessary models, thereby improving training efficiency.
>
> **8. Clarifications and Typo Corrections:**
>    We have redefined ambiguous variables in Algorithm 1 for clarity and corrected the identified typographical errors, including the expression "connecting preserving" and capitalization in Section 3.
>
> **9. Additional FL Baseline Performance Data:**
> Thank you for your suggestion to include additional FL baselines in Figure 3. However, our primary objective with Figure 3 is to emphasize the fast convergence achieved by our LSS method. The inclusion of FedAvg as a comparison baseline is intended to clearly demonstrate this aspect. Adding more baselines, while potentially informative, might not substantially contribute to this specific narrative focus.
>
> Your thorough review has been invaluable in enhancing the quality and clarity of our work. We are grateful for your time and effort, and we remain open to any further suggestions or discussions.

---

### Official Review · Reviewer_w5Zc · 2023-11-01

**Soundness:** 3 good
**Presentation:** 3 good
**Contribution:** 2 fair
**Rating:** 3
**Confidence:** 5

**Summary:**

This paper addresses the challenge of federated learning in the context of non-IID data distributions. Federated learning inherently leads to parameter heterogeneity across local devices, which may result in diminished model performance. To counteract this, the authors introduce a novel algorithm termed Local Superior Soups (LSS). The algorithm performs interpolation between randomly chosen local models during each local training phase. This interpolated model tends to be more stable, thereby enhancing its generalizability when integrated into FedAVG. For local training, two cost functions are also proposed, aimed at enhancing model diversity and affinity. These facilitate a more effective interpolation process and generally contribute to building a superior global model. Empirical results demonstrate the efficacy of the proposed algorithm.

**Strengths:**

1. The algorithm demonstrates superior performance compared to the state-of-the-art on CIFAR-10 and FMNIST benchmarks.

2. The methodology is easily understood.

**Weaknesses:**

1. The algorithm may necessitate significant memory resources for storing local training trajectories.

2. Discussion on computational overhead is lacking.

3. A more equitable comparison considering both computational and memory costs is needed.

4. Despite emphasizing the importance of communication costs for large models in the motivation, the experimental section does not incorporate such large pre-trained models.

5. The experiments are limited to relatively simple datasets, like CIFAR-10 and FMNIST; incorporation of more complex datasets such as CIFAR-100 is advised.

**Questions:**

See the Cons section for areas requiring further clarification or investigation.

---

> ### Author Response · Authors · 2023-11-21
> ****Response to Reviewer w5Zc****
>
> 1. **Memory and Computational Costs of Our Method:**
>    We acknowledge your concern about the memory and computational costs of our algorithm. In the revised manuscript, we have included a detailed analysis in the Appendix E.2 that compares the memory footprint of our method with other existing approaches. This analysis demonstrates how our approach effectively balances memory usage with computational efficiency.
>
> 2. **Inclusion of Large Pre-trained Models in Experiments:**
>   In the Appendix E.2 of the initial version, we have already included experiments that involve fine-tuning large models like Vision Transformer (ViT) with LORA. We will consider incorporating larger pre-trained models in the future work.
>
> 3. **Use of More Complex Datasets:**
>    The included experiments using the DomainNet dataset are more challenging than CIFAR-100, featuring higher resolution images and a larger variety of categories. This inclusion should provide a more comprehensive assessment of our algorithm's performance and applicability.
>
> We are grateful for your valuable suggestions. If further details or clarifications are needed on any of these points, please do not hesitate to let us know.

---

> > ### Comment · Reviewer_w5Zc · 2023-11-23
> >
> > Thank you for your response. I suggest revising the manuscript to include results for complex tasks using large pre-trained models as the motivation examples in Intro. Additionally, a more thorough discussion of the memory and computational complexity should be in the main text.

---

> > > ### Author Response · Authors · 2023-11-23
> > > **Response to Reviewer w5Zc**
> > >
> > > Thank you for your valuable suggestion. We understand the potential benefits of applying our method to larger-scale models and more complex datasets. However, as we state in the introduction (the firs pargraph the last sentence), **in this paper, we focus on reducing communication rounds in FL with a pre-trained model as initialization**. Our motivation example is intended to emphasize the increasing importance of using fewer communication rounds in FL as models become larger.
> > >
> > > While we acknowledge the significance of memory and computational overhead, these aspects are not the central metrics of focus in our study. Our primary emphasis is on demonstrating the effectiveness of our method in reducing communication rounds in FL settings. We have also elaborated in the main body and method section that arbitrarily increasing local steps cannot not reduce communication rounds effectively. And simply increasing model size only increases communication overhead.
> > >
> > > We would like to clarify that although our method may require additional memory during training, it does not increase the amount of parameter transferred to the server, as multiple local models are merged locally into one. This is a crucial aspect of our approach that maintains communication efficiency. Furthermore, our method allows for increasing local steps without causing significant client drift, thereby effectively reducing the number of necessary communication rounds.
> > >
> > > We hope this explanation provides clarity on the scope and focus of our study.

---

### Official Review · Reviewer_qnQ5 · 2023-11-01

**Soundness:** 3 good
**Presentation:** 2 fair
**Contribution:** 3 good
**Rating:** 6
**Confidence:** 4

**Summary:**

This paper proposes Local Superior Soups (LSS), which is an approach composed of several heuristic-based ingredients including random interpolation, diversity and affinity regularization. The approach aims to use model averaging to improve generalization, thus mitigating the non-iid issue in federated learning. The experimental results showcase the effectiveness of the proposed approach.

**Strengths:**

- The experimental results seem to obtain a non-trivial improvement across a range of baselines and benchmark datasets.
- The motivation of each component of the proposed approach is clear and intuitive.

**Weaknesses:**

- Presentation of Section 3, more specifically the theory part. In general, I feel there is some verbosity as well as disconnection with the rest of the paper. For example, the proposition 3.1, I understand the authors would like to convey $\tau$ has an upper bound thus can not be arbitrarily increased to decrease $R$. However, this seems never really referred to or validated in the rest of the paper.

- There are also several places unclear to me also in the theory part. For example, the authors claim "there is an additional error term in the convergence rate that monotonically increases with the number of local steps", which refers to the 3rd term of RHS. However, isn't the $\tau$ in the denominator?

And the main motivation of the model averaging is the authors claim the $\beta$, which is the Lipschitz constant, will be controlled by LSS. However, I do not see any validation of such connection except for some intuition-based argument. Thus, whether the logic can go through remains unclear to me.

- Based on my understanding, LSS replaces the multiple runs in model soups, which require several different hyperparameter specifications, with an iterative model averaging, i.e., average models continuously along with the optimization path. Basically, similarly to SWA? Currently the presentation of random interpolation is a bit confusing due to overcomplicated description.

- About the experimental setup, the authors describe using different local update steps for different approaches. I am not very clear about the reasoning. Is there any figure or table showing FedAvg cannot take more local steps? And how to ensure a fair comparison in this different local update regime.

**Questions:**

Please see weaknesses.

---

> ### Author Response · Authors · 2023-11-21
> ****Response to Reviewer qnQ5****
>
> 1. **Presentation and Connectivity of Section 3 (Theory):**
>   We thank you for your insightful feedback on the presentation of Section 3. In the revised manuscript, we have streamlined the section to reduce verbosity and enhance its connection with the rest of the paper. We have also moved some detailed analyses to Appendix B.2 to maintain the flow while ensuring the availability of in-depth information.
>
> 2. **Relevance and Validation of Proposition 3.1:**
>   We acknowledge your observation regarding the lack of validation of Proposition 3.1. To address this, we have included an empirical verification in Appendix E.2 concerning FL methods for reducing communication rounds, which cannot be achieved simply by increasing local epochs.
>
> 3. **Clarification on Additional Error Term in Convergence Rate:**
>   Thanks for your correction. In the revised version, Section 3.2 has been updated to provide a clearer explanation and the correct error term index.
>
> 4. **LSS Smoothness:**
>   In response, we have included empirical studies in Appendix E.2. These studies demonstrate the effect of LSS on the improved smoothness and provide evidence to support our theoretical claims.
>
> 5. **Comparison with Stochastic Weight Averaging (SWA) and Random Interpolation:**
>   To clarify the differences with SWA and the concept of random interpolation, we have revised Section 3.3. This includes a more straightforward and less complicated description of our methodology, differentiating it from existing techniques like SWA.
>
> 6. **Experimental Setup and Rationale for Different Local Update Steps:**
>   To address this, we have added new empirical evidence in Appendix E.2, showcasing the reasoning behind choosing different local steps for FedAvg.
>
> We deeply appreciate the time and effort you have put into reviewing our work. Your feedback has been instrumental in enhancing the clarity and rigor of our manuscript. If any part of our response requires further elaboration or discussion, please let us know.

---

### Meta-Review · Area_Chair_d3YF · 2023-12-11

**Metareview:**

This paper proposes an approach Local Superior Soups (LSS) which uses several heuristics including random interpolation, diversity and affinity regularization to improve the generalization performance in FL settings. The authors provide empirical evidence to demonstrate the efficacy of the proposed algorithm.

**Justification For Why Not Higher Score:**

I agree with the reviewers that the presentation & organization of the paper is very poor and there is lack of rigor in the approach. Also the paper would benefit from more comprehensive experiments.  While the authors addressed a few of the concerns in their response, it was not satisfactory to the reviewers.

**Justification For Why Not Lower Score:**

N/A

---

### Decision · Program_Chairs · 2024-01-16

Reject